# Site-Specific Antibody Conjugation with Payloads beyond Cytotoxins

**DOI:** 10.3390/molecules28030917

**Published:** 2023-01-17

**Authors:** Qun Zhou

**Affiliations:** Biologics Innovation, Large Molecules Research, Sanofi, Cambridge, MA 02141, USA; qun.zhou@sanofi.com; Tel.: +1-617-685-7243

**Keywords:** site-specific antibody conjugation, engineering, payloads, siRNA, degraders, peptides/proteins

## Abstract

As antibody–drug conjugates have become a very important modality for cancer therapy, many site-specific conjugation approaches have been developed for generating homogenous molecules. The selective antibody coupling is achieved through antibody engineering by introducing specific amino acid or unnatural amino acid residues, peptides, and glycans. In addition to the use of synthetic cytotoxins, these novel methods have been applied for the conjugation of other payloads, including non-cytotoxic compounds, proteins/peptides, glycans, lipids, and nucleic acids. The non-cytotoxic compounds include polyethylene glycol, antibiotics, protein degraders (PROTAC and LYTAC), immunomodulating agents, enzyme inhibitors and protein ligands. Different small proteins or peptides have been selectively conjugated through unnatural amino acid using click chemistry, engineered C-terminal formylglycine for oxime or click chemistry, or specific ligation or transpeptidation with or without enzymes. Although the antibody protamine peptide fusions have been extensively used for siRNA coupling during early studies, direct conjugations through engineered cysteine or lysine residues have been demonstrated later. These site-specific antibody conjugates containing these payloads other than cytotoxic compounds can be used in proof-of-concept studies and in developing new therapeutics for unmet medical needs.

## 1. Introduction

Antibody–drug conjugation has gained significant momentum during the past few years with more than ten antibody–drug conjugates (ADCs) being approved by regulatory agencies for cancer treatment in clinics [1,2,3,4,5]. As hybrid molecules containing biologics and highly toxic low-molecular weight chemotherapeutic drugs, ADCs leverage the advantages of both targeting specificity of antibodies and high potency of cytotoxic compounds or synthetic cytotoxins. To synthesize ADCs, the antibodies are coupled with drug-linkers using different conjugation chemistries. The therapeutic index of the ADCs depends on many attributes including the expression profiles of selected cancer antigens, the qualities and specificities of antibodies, the properties of the synthetic cytotoxins (potency, mechanism of action, loading, cleavable or non-cleavable linkers), and the conjugation chemistries used [6]. The conventional conjugation approaches rely on non-specific/stochastic coupling of drug-linkers to lysines (about 40 residues per IgG1) or hinge cysteines (8 residues per IgG1). They often result in a heterogeneous profile of ADCs with a drug-to-antibody ratio (DAR) of 2 or 4, leading to difficulties in characterization and process control. To overcome these disadvantages, next generation site-specific antibody–drug conjugation methods have been developed. These methods have been reviewed in many excellent publications [7,8,9,10,11,12,13].

In addition to using synthetic cytotoxins, there is increased interest in coupling other payloads with site-specific antibody conjugation. These payloads include non-cytotoxic compounds that are not cytotoxic to human cells, as well as proteins/peptides, glycans, lipids, and nucleic acids. The review herein highlights the progress in site-specific conjugation of these payloads other than synthetic cytotoxins to antibody molecules after presenting a brief overview of advances in developing next generation antibody conjugation methods.

## 2. Overview of Site-Specific Antibody Conjugation

Site-specific antibody conjugation begins with the engineering or modification of monoclonal antibody, followed by the conjugation of optimized drug-linkers (Figure 1). The antibody is engineered through the Fab or Fc region of an IgG to introduce different conjugation sites by using genetic engineering, metabolic labeling or chemoenzymatic modification. Many different methods can be categorized through different conjugation sites in the antibody (Table 1, Figure 1).

As described in many site-specific antibody conjugations, genetic engineering has been carried out by introducing specific sites for conjugation, such as cysteine (Cys), glutamine (Gln), unnatural amino acids (*p*-acetylphenylalanine or pAcF and *p*-azidomethyl-L-phenylalanine or pAMF), or short peptide tags. The mutated DNA is transfected into the cells to express engineered antibody. Cell line engineering is often required for the genetic engineering if the antibody of interest would be used for therapeutic developments. Once it is established, the conjugation process is straightforward. Metabolic labeling is applied for several methods, such as site-specific conjugation through unnatural amino acids, and its selectivity and efficiency have been demonstrated [24,25]. Since progress in site-specific antibody conjugations has been reviewed in detail previously [7,8,9,10,34,43,44], only the recent advances are highlighted here.

The conjugation through Cys is relatively simple and straightforward without the need for special reagents or enzymes, although there is potential instability in vivo, depending on location of the engineered Cys residue. THIOMAB^TM^, the site-specific conjugation through engineered Cys, was one of the first site-specific antibody drug conjugation methods being developed [14]. The ADC generated using this method shows not only high homogeneity, but also increased efficacy and therapeutic index in vivo in animal models. There are many different sites in Fab and Fc regions that have been engineered to introduce single unpaired Cys residues for site-specific conjugation [15,45,46,47,48,49]. Recently, antibody engineering by introducing double and triple unpaired Cys has also been described leading to conjugates with more payloads (DAR of more than two) per antibody [22,50,51,52,53].

The unnatural amino acid can be introduced at different positions in an antibody providing the sites for stable conjugation. However, special reagents and extensive cell line engineering are often required for the approach.

Conjugations through chemoenzymatic modification methods have been demonstrated by using multiple enzymes including transglutaminase, transpeptidase sortases, glycosyltransferases or endoglycosidases. There is no need for genetic engineering with the conjugation through antibody glycans, but special reagents and enzymes are necessary. Enzymatic modifications of amino acids within specific short peptide tags generate antibody conjugates coupled with payloads at high selectivity and stability although the potential immunogenicity of introduced peptide tags is currently unknown. Although chemoenzymatic methods have been demonstrated as efficient processes, the reagents and associated cost may need to be considered for process scale-up. It was recently reported that the introduction of small-molecule drugs into Gln of an aglycosylated antibody by using microbial transglutaminase (mTG) can have an impact on the stability of ADC. The hydrophobic cytotoxin was able to compensate for thermal destabilization resulting from structural distortions due to antibody deglycosylation [21]. The site-specific conjugation of Q295 in deglycosylated antibody was also described by using the same transglutaminase with cystamine for thiolation [23]. The chemically introduced thiol on Q295 could be selectively conjugated using maleimide chemistry with improved plasma stability. In another report, Wijdeven et al. described an improved method from enzymatic glycan remodeling followed by metal-free click chemistry [35]. An engineered endoglycosidase and a native glycosyltransferase were selected for chemoenzymatic reaction using a novel azido sugar, generating ADCs with improved efficacy. Recently, there were advances from site-specific antibody conjugation using transglycosylation. A one-pot reaction for deglycosylation and transglycosylation in antibody Fc was described using wild-type endoglycosidase from *Streptococcus pyogenes* of serotype M49 (Endo-S2) [54,55]. The enzyme was shown to efficiently introduce the functionalized disaccharide oxazolines carrying site-selectively modified azide in varied numbers, resulting in ADCs with a precise control of DAR ranging from 2 to 12 via a copper-free strain-promoted click chemistry. Endo-S2 was able to accommodate drug-preloaded minimal disaccharide derivative oxazolines as donor substrates for efficient transfer of the glycan containing drug-linker. These ADCs containing monomethyl auristatin E (MMAE) with higher DARs were shown to be more potent in killing antigen-overexpressing cancer cells than those with lower DARs. The in vivo anticancer efficacy in tumor xenograft model was reported with MMAE-conjugated ADCs generated using a similar approach [56].

Among many different methods as described, the site-specific conjugations through engineered Cys, unnatural amino acid, and enzymatic glycan remodeling–metal-free click chemistry have been performed at big scale and the produced conjugates are being tested in clinical trials [11,35]. Besides ADCs, site-specific antibody conjugation methods have also been applied for coupling other payloads as described in following sections (Figure 1).

## 3. Non-Cytotoxic Compounds as Payloads

There are several non-cytotoxic compounds used for selective conjugation (Table 2). They include polyethylene glycol (PEG), antibiotics, immuno-modulating compounds, protein degraders, and ligands for receptors and proteins that are overexpressed in cancer.

### 3.1. PEG

To prolong the serum half-life of antibody fragments, the PEGylation of Fab through hinge Cys residues was first described more than 20 years ago [71]. Later research on Fab engineering led to PEGylation at introduced unpaired Cys residue at the C-terminal end of the heavy chain (HC) constant region 1 [57,58,59]. Since the attachment of PEG (20 to 40 kDa in size) is far away from the epitope binding region, the antigen binding and in vitro bioactivity of PEGylated Fab are usually not reduced as compared to unmodified Fab. The PEGylation also does not affect stability, while the half-life of the antibody fragment can be significantly increased. A PEGylated anti-TNF Fab, Certolizumab pegol, was approved by regulatory agencies for treating patients with immune-mediated inflammatory diseases, while other PEGylated Fab constructs are under proclinical and clinical developments. It would be interesting to compare pharmacokinetics and pharmacodynamics of PEGylated Fab, which is monovalent and has no effector function, with the bivalent antibody IgG.

### 3.2. Antibiotics

A novel antibody–antibiotic conjugate (AAC) was described as a potential therapeutic that effectively kills intracellular bacteria [60]. The anti-*S. aureus* antibody was cloned and purified from B cells derived from the peripheral blood of patients recovering from various *S. aureus* infections. It was selected against wall-teichoic acids, pathogen-specific polyanionic glycopolymers that are connected to the thick peptidoglycan layers of gram-positive bacteria. A highly efficacious antibiotics rifalogue, which is activated only after being released in lysosomes, was conjugated to the antibody engineered with V205C at the light chain (LC) using the THIOMAB^TM^ approach. The AAC is superior to vancomycin for treatment of bacteremia in vivo and elimination of intracellular *S. aureus* infections. It substantially reduced bacterial load in the heart, kidney, and bones from mice on 7 and 14 days after a single intravenous administration [61]. The AAC provides a unique therapeutic approach against intracellular bacterial infection, and it is currently under clinical developments (ClinicalTrials.gov Identifier: NCT03162250).

### 3.3. Immune-Modulating Compounds

Synthetic non-cytotoxic enzyme inhibitors or ligands for specific receptors have been coupled to antibodies through site-specific conjugations for immune modulation. An enzyme inhibitor to phosphodiesterase 4 (PDE4) has been utilized as a payload [62]. Although many PDE4 inhibitors have demonstrated anti-inflammatory activities, some of them showed dose-limiting side effects. To exploit tissue-restricted delivery of these inhibitors for increasing therapeutic index, a highly potent PDE4 inhibitor was coupled to an anti-CD11a, a surface antigen highly expressed by leukocytes including myeloid cells and lymphocytes. The antibody, which was engineered with the unnatural amino acid, pAcF, at A122 of the heavy chain as well as other mutations to silence effector functions, was selectively conjugated with aminooxy containing inhibitor using oxime chemistry. The immunoconjugate was rapidly internalized into immune cells and suppressed lipopolysaccharide-induced TNFα secretion in primary human monocytes. It displayed an in vivo anti-inflammatory effect in mouse models. The myeloid-cell-specific anti-Ly6C/G VHH molecules were also conjugated with dexamethasone, which contains an acid-labile hydrazone moiety, using sortase A (SrtA) mediated transpeptidation [64]. The conjugates enabled specific delivery of the dexamethasone, which has undesirable side effects, onto bronchial epithelium in influenza virus-infected mice. The VHH conjugates, but not free dexamethasone, reduced the weight loss of animals infected with virus. Several liver X receptor (LXR) agonists were investigated as a potential therapy for diseases including atherosclerosis based on their ability to induce reverse cholesterol transport and anti-inflammation [63]. However, they induced excessive lipogenesis in liver through their interaction with LXR-α. To prevent the on-target adverse effect, the LXR agonist was coupled to anti-CD11a, which was selected for targeting since the protein is highly expressed in macrophage and monocytes but not hepatocytes. The increased expression of CD11a on monocytes has been found to be correlated with atherosclerotic coronary stenosis [72]. An unnatural amino acid, pAcF, was introduced at position A122 of the antibody’s HC and conjugated with an aminooxy containing LXR agonist with a cleavable linker sensitive to lysosomal cathepsin B. The immunoconjugate induced LXR activation specifically in human THP-1 monocyte and macrophage cells in vitro, while it had no significant effect in hepatocytes. It would be interesting to see more research with antibody conjugated with immune modulating compounds being tested in different disease models.

### 3.4. Protein Degraders: PROTAC

Recently, the site-specific antibody conjugation was applied to the field of protein degradation. The concept of proteolysis-targeting chimera (PROTAC) was first introduced more than 20 years ago [73,74]. It relies on synthetic chimeric molecule for intracellular degradation of protein target or protein of interest (POI) through the proteasome (Figure 2). The PROTAC degraders often contain three components: a ligand that binds to the POI, a ligand for E3 ubiquitin ligase, and a space group or linker that connects the first two components. During action, the PROTAC degrader, except for molecular glues, forms a ternary complex among itself, a POI, and an E3 ubiquitin ligase that leads to ubiquitination of the target for subsequent destruction via trafficking to the proteasome. The PROTAC technology makes it feasible for effective degradation of many intracellular protein targets, such as tyrosine kinases, hormone receptors, and transcription factors which are often undruggable via many conventional inhibitors. It also provides different mechanism of actions (event-driven pharmacology) as compared to conventional inhibitors (occupancy-driven pharmacology) [75]. Its potential to expand the application in drug discovery has generated great interests, resulting in significant progress during the last decade with several PROTAC degraders reaching clinical trials [75,76]. However, there are still some challenges associated with physico-chemical properties of some degraders such as relatively large entities that may compromise oral bioavailability, solubility, and/or in vivo pharmacokinetics [53]. To provide an alternate approach to administering chimeric degrader compounds in vivo, significant efforts have been made to generate antibody-PROTAC conjugates using site-specific conjugation approaches.

Several PROTAC compounds have been identified for degradation of bromodomain-containing protein-4 (BRD4), a member of the bromodomain and extra-terminal (BET) family of protein, that functions as an epigenetic “reader” of acetylated histone lysine residues [66]. Although the lead degraders displayed extremely potent BRD4 degradation in vitro, several compounds exhibited unfavorable physiochemical characteristics and poor in vivo pharmacokinetic properties following intravenous or oral administration to mice. Therefore, the degrader was conjugated to an antibody against C-type lectin-like molecule-1 (CLL1 or CLEC12A), which was engineered with three unpaired cysteine residues (LC-K149C, HC-L174C, and HC-Y373C), through methanethiosulfonyl-Cys conjugation. Despite the relatively high lipophilicity of the PROTAC-linker, it was still possible to attach six degraders per antibody (drug-antibody ratio or DAR of ~6). A single intravenous administration of the antibody–degrader conjugate led to dose-dependent tumor growth inhibition in mice bearing HL-60 (acute myeloid leukemia) cells xenografts. In another study, other BRD4 degraders were conjugated by the same laboratory to an antibody against six transmembrane epithelial antigen of the prostate 1 (STEAP1) overexpressed in prostate cancer [51]. The antibody was introduced with either a single Cys (LC-K149C) or three unpaired Cys (LC-K149C, HC-L174C, and HC-Y373C) and conjugated with BRD4 degraders with DAR of ~2 and ~6, respectively, using thiol-maleimide chemistry. The conjugates exhibited intracellular delivery of the payloads to PC3-S1 prostate cancer cells along with reductions in intracellular BRD4 levels and in MYC transcription in vitro. The same laboratory identified additional potent BRD4 degraders and converted them to protease-cleavable linkers bearing methanethiosulfonate functionality [52]. These degrader-linkers were coupled to the antibody against STEAP1 engineered with three unpaired Cys as described above with DAR of close to 6. The conjugates exhibited highly potent BRD4 degradation and antiproliferation activity against prostate cancer cell line PC3-S1 in vitro. The antibody conjugates bearing BRD4 degraders also showed strong anticancer efficacy in vivo in mouse xenograft assessments that employ several different cancer models. For selective protein degradation in breast cancer cells, a BRD4 degrader was conjugated to anti-HER2 (trastuzumab) by a different laboratory [65]. The antibody-PROTAC conjugate was prepared by using a novel dibromomaleimidestrained alkyne linker to rebridge the partially reduced interchain disulfide bonds and then coupling the protein degrader using copper-free strain-promoted azide–alkyne cycloaddition. The conjugate was internalized, resulting in active PROTAC for BRD4 degradation only in HER2 positive breast cancer cell lines in vitro. Furthermore, the field on selective degradation of estrogen receptor has also been advanced [75]. The site-specific antibody conjugation was described by using degraders that target estrogen receptor alpha (ERα) [50,77]. Anti-HER2–PROTAC conjugates containing ERα degraders were prepared using the same methods as described above by coupling selectively to engineered Cys residues [50]. They showed reasonably favorable in vivo stability and the degrader payloads were efficiently released after internalization in vitro.

The site-specific antibody conjugation with PROTAC provides an alternative approach to administer chimeric degrader compounds in vivo or to deliver them into specific cells or tissues. It may be very useful for PROTACs with sub-optimal drug properties and pharmacokinetics or need in cell/tissue specific delivery.

### 3.5. Protein Degraders: LYTAC

In addition to PROTAC, site-specific conjugation has also been applied for lysosome-targeting chimeras (LYTAC) [78]. LYTAC relies on the interaction of ligands with its lysosomal-targeting receptor driving the degradation of extracellular protein targets or proteins of interest (POI), into the lysosome for degradation. It contains two parts: one is the ligand for lysosomal-targeting receptor while the other is the antibody against extracellular POI. Banik et al. prepared LYTAC by performing non-specific conjugation of antibodies against multiple extracellular POI with mannose-6-phosphonate polymer that binds to cation-independent mannose-6-phosphate receptor (CI-MPR) [79]. They showed efficient degradation of soluble proteins (APOE4) and membrane proteins (EGFR, CD71 and PD-L1) in cell lines in vitro. Recently, site-specific antibody conjugation was reported for the generation of LYTAC using chemoenzymatic methods [68]. Antibodies against HER2 (trastuzumab) and EGFR (cetuximab) were modified using endoglycosidase S for the deglycosylation and transfer of a synthetic high-affinity mannose-6-phosphate (M6P) glycan oxazoline, generating antibody-M6P glycan conjugates. The M6P containing LYTACs were able to selectively degrade membrane HER2 and EGFR in vitro. In addition to the CI-MPR, the degradation of POI with LYTAC binding to another lysosomal-target receptor, the asialoglycoprotein receptor (ASGPR), was also demonstrated [67,80]. Ahn et al. developed LYTACs that engage ASGPR, which is only expressed in hepatocytes, to degrade extracellular soluble and membrane proteins [67]. Antibodies against extracellular protein targets were conjugated with a triantenerrary N-acetylgalactosamine (GalNAc) that binds ASGPR in high affinity. A reactive aldehyde handle from formylglycine (FGly) was generated by FGly generating enzyme (FGE)-catalyzed oxidation of a specific Cys residue from an introduced short peptide tag, LCTPSR, in the antibodies. It was then conjugated to triantenerrary GalNAc using a hydrazine-iso-Pictet–Spangler reaction and copper-free click chemistry [81]. The antibody-GalNAc conjugate displayed efficient degradation of membrane receptors EGFR and HER2 on hepatocellular carcinoma cells, such as HepG2 cells, in vitro. The work has provided an example of targeted protein degradation in specific tissue such as liver.

### 3.6. Ligand for Proteins Overexpressed in Cancer

There are reports on site-specific conjugation of antibodies with ligands for proteins overexpressed in cancers [69,70]. The anti-CD3 Fab or single-chain variable fragment antibody (scFv) was coupled to these ligands for the generation of bispecific T-cell engagers. These bispecific T-cell engagers are unique in that an anti-CD3 Fab or scFv is coupled with only a synthetic compound as ligands but is not linked to another antibody against cancer-associated proteins. In one study, an unnatural amino acid, pAcF, was incorporated into two different locations (LC-S202 and HC-K138) of the anti-CD3 Fab that are distal to antibody paratope based on its crystal structure [69]. The engineered Fab was conjugated to a synthetic high-affinity ligand for prostate-specific membrane antigen (PSMA). The bispecific T-cell engager showed potent cytotoxicity against prostate cancer cell lines in vitro and strong anticancer efficacy in vivo. In another study, high-affinity ligands for folate receptor α and integrin α4β1, which are overexpressed in multiple cancer cells, were coupled into introduced C-terminal selenocysteine (Sec) of an engineered anti-CD3 Fab [70]. The bispecific antibody–synthetic compound conjugates displayed potent cytotoxicity in vitro and ex vivo against cancer cell lines and primary cancer cells in the presence of T cells. They may be useful for screening or designing better biologics if the synthetic ligands are available.

Chemically programmed antibodies (cpAbs) have been described for almost twenty years [69,82,83,84]. They were generated by using site-specific and covalent conjugation of small molecules to non-targeting mAbs with unique reactivity centers, such as an antibody against 1,3-diketone that was generated using reactive immunization. A unique reactive (usually nucleophilic) residue, such as lysine, in the antibody was coupled to synthetic compounds or peptides with a reactive (usually electrophilic) group. In the case of the anti-1,3-diketone antibody, the reversible covalent interaction of the lysine residue in the reactivity center with the 1,3-diketone forms an enaminone stabilized by an imine-enamine tautomerism. The cpAbs rely on coupled synthetic compounds for the recognition of extracellular antigens. Walseng et al. reported how a diabody containing both anti-hapten and anti-CD3 Fv (disulfide-linked polypeptides containing either VH or VL) can be coupled with hapten-derivatized folate through a reactive lysine introduced in one of the polypeptides of anti-hapten antibody [85]. The chemically programmed diabody demonstrated high selectivity and potency against folate receptor α-expressing ovarian cancer cells both in vitro and in vivo. The work is very interesting, but the potential immunogenicity of those constructs in human is unknown.

Antibodies against HIV gp41 were engineered for site-specific conjugation with cholesterol [86]. The unpaired cysteine residues were introduced into antibodies as T20C on VL and S444C on HC for conjugation with maleimide containing cholesterol. It was demonstrated that the antibody–cholesterol conjugate could rescue antiviral activity of a mutant of a broadly neutralizing anti-HIV antibody with hydrophobic CDR H3 loops. The cholesterol component provided enrichment of the conjugate in lipid raft of the plasma membrane, facilitating recognition of protein epitope from the membrane-proximal external region of HIV gp41. The antibody conjugate also increased the antiviral activity of that wild-type antibody as well as another non-membrane-binding HIV antibody.

Antibody conjugates containing non-cytotoxic compounds, which have been generated using site-specific conjugation, demonstrated impressive in vitro and in vivo results. It may be interesting to compare them with other antibody formats, such as those being generated using recombinant approaches, if it is feasible, in proof-of-concept studies. Nevertheless, they have shown potential as unique therapeutics for unmet medical needs.

## 4. Proteins or Peptides as Payloads

Besides the non-cytotoxic compounds, proteins or peptides have also been conjugated to antibodies using site-specific approaches. Among them, bioconjugation for making bispecific antibodies was described a long time ago before new generation of recombinant approaches were available [87]. With many novel methods being developed, site-specific conjugations have been applied for coupling proteins or peptides to antibodies (see Table 3).

### 4.1. Proteins

Several antibody–protein conjugates were produced and characterized for different applications [42,88,89,90]. A novel antibody–sialidase conjugate was reported [88]. A short peptide tag, LCTPSR, was introduced near the C-terminus of the anti-HER2 antibody heavy chain. An unusual aldehyde-bearing formylglycine (FGly) was generated through the oxidation of cysteine in the consensus sequence by co-expressed FGly generating enzyme (FGE) before it was converted to an azide using a heterobifunctional linker aminooxy-tetraethyleneglycol-azide. The azide containing antibody was coupled to a bicyclononyne modified sialidase from *V. cholerae* through click chemistry. The antibody–sialidase conjugate showed enhanced ADCC activity against HER2 positive cancer cells. The increased activity was correlated with cancer cell desialylation, reduced binding by nature killer cells (NK) to inhibitory sialic acid-binding Ig-like lectin (Siglec) receptors, and enhanced binding to NK-activating receptor natural killer group 2D. Recently, a new study, which was reported by the same laboratory from Bertozzi et al., demonstrated the anticancer immune response of antibody–sialidase conjugates in vivo [89]. To reduce antibody-independent activity, sialidase from *Salmonella typhimurium* with relatively high *K*_M_ value was chosen for preparing antibody–sialidase conjugate. The conjugate was generated using similar click chemistry as described above except that a different heterobifunctional linker, hydrazine-iso-Pictet–Spenger-azide, was used to enhance stability, and sialidase was modified with an α-chloroacetamide-dibenzocyclooctyne linker at an engineered C-terminal Cys residue. The antibody–sialidase conjugate with about one sialidase per antibody displayed anticancer activity with prolonged mouse survival in syngeneic breast cancer models. It desialylated breast cancer cells, leading to enhanced immune cell infiltration that depends on a Siglec-E checkpoint receptor found on tumor-infiltrating myeloid cells. The concept of work led to design and generate a recombinant version by others for preclinical and clinical development (ClinicalTrials.gov Identifier: NCT05259696).

Peptide-directed photo-cross-linking for site-specific antibody conjugation has also been developed [42]. Antibody against HER2 was conjugated with an engineered *Pseudomonas* exotoxin A, a bacterial toxin containing Fc-binding peptide that includes a photoreactive amino acid analogue, *p*-benzoylphenylalanine. Although it was difficult to express the antibody-exotoxin fusion protein in bacterial or mammalian cells, the conjugation approach resulted in high yield for monoconjugated form after exposure to UV light. The antibody-bacterial toxin conjugate exhibited potent cytotoxicity toward HER2-positive cancer cell lines.

Transpeptidation reaction mediated by bacterial sortases has been applied in producing site-specific antibody-protein/peptide conjugates. Among many different sortases, sortase A (SrtA) from *Staphylococcus aureus* has been used frequently in the transpeptidation [99]. It uses an LPXTG as substrate generating a thioester-linked acyl enzyme intermediate that is intercepted by an aminoglycine nucleophile containing acceptor. The reaction leads to site-specific ligation of an acyl donor (antibody containing LPXTG sequence in C-terminus of heavy chain) and acceptor (proteins/peptides containing N-terminal Gly). SrtA-mediated transpeptidation method was used for site-specific antibody-protein conjugation [90]. Two mutant IgG antibodies of anti-HER2 (trastuzumab) and anti-EGFR (cetuximab), which contain residues LPSTGGK at C-terminus, were coupled with the side chain of anthrax toxin protective antigen (PA) containing three Gly residues in N-terminus. The antibody conjugates displayed significant intracellular delivery through PA-mediated oligomerization and translocation. They were able to deliver edema factor and N-terminus of lethal factor (LFN) fused with diphtheria toxin and Ras/Rap1-specific endopeptidase into cytosol of cancer cell lines and showed strong cytotoxicity with diphtheria toxin.

The conjugations using bacterial toxin for indirect cytotoxicity or intracellular delivery have provided different examples from those using low-molecular weight synthetic cytotoxins. The research is interesting but the potential immunogenicity of these constructs in human and its impact on pharmacokinetics and pharmacodynamics are currently unknown.

### 4.2. Antibody Fragments

Different antibody fragments have also been conjugated with each other using site-specific method generating bispecific antibody Fab conjugates [91]. A tRNA/aaRS pair derived from *M. jannaschii* was co-expressed to incorporate an unnatural amino acid, pAcF, at defined sites in each of two Fab fragments, anti-CD3 (HC-K138) and anti-HER2 (LC-S202) in response to an amber nonsense codon. The two Fabs containing the pAcF residues were conjugated with heterobifunctional linkers containing either an azide or cyclooctyne group before they were coupled using a copper-free click chemistry. The bispecific T-cell engager containing two different Fab fragments efficiently recruited T cells from human peripheral blood mononuclear cells (PBMCs) to kill cancer cell targets at picomolar concentration.

A similar site-specific conjugation method was applied to prepare another bispecific antibody Fab conjugates [92]. The Fabs against CD3 and C-type lectin-like molecule-1 (CLL1) were coupled to azido-PEG3-aminooxy and BCN-PEG3-aminooxy linkers, respectively, before they were conjugated using click chemistry. A different bispecific antibody Fabs against CD3 and CD33 were also prepared using the same method. The CLL1 and CD33 are proteins overexpressed in acute myeloid leukemia. The bispecific antibody Fab conjugate against both CD3 and CLL1 displayed strong in vitro and in vivo anticancer activities as compared to the bispecific Fab conjugate against CD3 and CD33.

In addition to the use of PEG linker for coupling in generating bispecific antibody Fabs, either oligonucleotides or peptide nucleic acids of defined sequences were site-specifically coupled to unnatural amino acid introduced in antibody for preparation of bispecific T cell engager [93]. As described above, pAcF was incorporated into HC-K138 of anti-CD3 Fab and LC-S202 of anti-HER2 Fab, respectively. Complementary peptide nucleic acid strands were then coupled to both Fab fragments. The bispecific Fab conjugates were self-assembled based on Watson–Crick base pairing properties of oligonucleotides. The bispecific T cell engagers were able to recruit cytotoxic T cells to kill cancer cells in vitro. Tetrameric Fab conjugates were also generated using similar approach. The SpyTag/SpyCatcher system has been applied for biparatopic antibody conjugation [94,95]. The technology for irreversible conjugation of recombinant proteins was developed ten years ago [100] with improvement over the years [101]. The conjugation relies on spontaneous reaction between peptide SpyTag (~13 amino acids) and the protein SpyCatcher at low nanomolar concentration to form an intermolecular isopeptide bond between the pair. Reaction occurs in high yield within minutes leading to irreversible covalent linked fusion protein and peptide as a valuable tool for studying protein-protein interaction. The SpyTag/SpyCatcher system has been used for generating antibody fusions [95]. Two different scFv antibody fragments that bind different domains of a cancer-related antigen, roundabout homolog 1 (Robo1), were fused to SpyTag and SpyCatcher at C-terminus. The binding of these two scFv fusion proteins to different domains of the antigen was demonstrated by using bio-layer interferometry. Although the antibody fused with large nonhuman peptides could potentially lead to immunogenicity in human, the system could be useful for high-throughput screening of different bispecific or biparatopic antibodies.

The site-specific conjugation of two different antibody fragments have generated bispecific T cell engagers which showed anticancer activities. It would be interesting to compare them with those produced using recombinant approach that is relatively cost-effective. Nevertheless, the method may be useful for screening and designing optimal molecules.

### 4.3. Peptides or Cyclic Peptides

Touti et al. reported the site-specific conjugation of antibody with bactericidal macrocyclic peptide generating antibody-bactericidal conjugates (ABCs) [96]. An antibacterial antibody against lipopolysaccharide was fused with SrtA acceptor LPETG peptide in the C-terminus of heavy chain. The N-terminal Gly_3_ perfluoroaryl macrocyclic antimicrobial peptides, which were identified from a library with non-hemolytic and serum stable properties, were then conjugated to the antibody using SrtA. The ABC showed activity at nanomolar concentrations against *E. coli*. In another study, anti-EGFR VHH containing C-terminal fusion of LPETG peptide was conjugated with a Cell-Penetrating Peptide (CPP) using SrtA-mediated transpeptidation [97]. The VHH conjugate containing the CPP, a peptide corresponding to amino acids 38–59 of the human milk protein lactoferrin, showed receptor internalization and blocks EGFR activation.

Finally, expressed protein ligation through internal protein segment (intein), which is involved in protein splicing, was applied for antibody-peptide conjugation [98]. VHH against GFP was fused with Ala_3_-intein in C-terminus and it reacted with a cyclic arginine-rich cell-penetrating peptide (CPP), cyclic R_10_ that associates strongly with the nucleolus and contains Cys at N-terminus. The VHH-CPP conjugates were produced at good conversion rates (above 70%). It was demonstrated the intracellular delivery of these VHH constructs and relocation of proliferating cell nuclear antigen (PCNA) and tumor suppressor p53 when these recombinant proteins were tagged with GFP. The VHH constructs were able to co-transport GFP or GFP-PCNA and GFP-Mecp2 into cells in vitro (2–10% of cells detected).

The site-specific antibody–protein/peptide conjugations are useful alternate methods for production when the fusion proteins are difficult to be produced by using recombinant approaches with low yield or sub-optimal post-translational modification. They may provide constructs quickly for proof-of-concept study or screening. It could be valuable to compare the antibody–protein/peptide conjugates with antibody protein/peptide fusions in vitro and in vivo if both formats are available.

## 5. Nucleic Acid as Payloads

The therapeutic use of nucleic acids, including oligonucleotides, has been clinically validated with different mechanisms of action. They include nusinersen, an antisense oligonucleotides for treatment of spinal muscular atrophy through intronic splice modulation, and patisiran, a short interfering RNA (siRNA) for the treatment of ATTR amyloidosis through mRNA silencing [102]. The siRNA is a double-stranded 21–25-nucleotide complex which is highly selective in inducing degradation of a particular mRNA and inhibiting its translation. To enhance the specific tissue or cell delivery, many efforts have been made in developing site-specific conjugations in generating antibody–siRNA conjugates (ARCs) [103,104,105]. The ARCs displayed specific delivery into tissues, virus-infected cells, or cancer cells in vitro and in vivo. When it binds to a specific receptor on cell surface, the ARC is internalized into endosome inside the cells (Figure 3). After endosomal escape, siRNA is first loaded into a multiprotein RNA-induced silencing complex (RISC). The one of double-stranded siRNA, the passenger strand, is ejected from the RISC and the other, guide strand, is then complexed with a specific mRNA, resulting in cleavage of that transcript. The ARCs have been generated by using either noncovalent hybridization or direct conjugation with oligonucleotides to antibody or antibody fragments (Fab, scFv) (Table 4).

### 5.1. Non-Covalent Complex of siRNA with Peptide or Polymer

Song et al. first described the method using non-covalent complex for delivering siRNA [106]. The C-terminus of the heavy chain from an anti-gp160 (HIV envelope) Fab fragment was fused using a recombinant approach with the protamine, a small protein containing many positively charged amino acid residues interactive with DNA in sperm. The protamine from Fab fusion was then hybridized with siRNA, which was delivered and silenced gene expression specifically in cells expressing HIV-1 envelope and inhibited HIV replication in hard-to-transfect, HIV-infected primary T cells. The siRNAs were only delivered into HIV gp160 expressing B16 melanoma cells by intravenous or intratumoural injection into mice of the Fab-protamine fusion-complexed siRNA. An HER2 single-chain antibody (scFv) fused with protamine also delivered siRNA specifically into HER2-positive breast cancer cells. There are more reports on the uses of protamine fused antibody or antibody scFv for gene knockdowns in vitro or in vivo [108,109]. In one study, a scFv fragment from antibody against hantavirus surface envelope glycoprotein was fused with truncated protamine [109]. The scFv-protamine fusion protein was then complexed with siRNA that targets the encoding sequence of hantavirus and inhibited its replication. The antibody scFv-siRNA conjugate was specifically delivered into virus-infected cells and efficiently inhibited viral replication. It also displayed efficacy in reducing antigen levels of virus in mice in vivo and effectively protecting from viral infection-derived animal death in an encephalitis mouse model. In another study, scFv of antibody against human integrin lymphocyte function-associated antigen-1 (LFA-1) has been fused at C-terminus with basic peptide from human protamine (amino acid 8–29) [108]. The fusion protein was complexed with siRNAs and selectively delivered these siRNAs into activated leukocytes and silenced the expression of several genes in vitro, including those encoding Ku70, CCR5, and cyclin D1. It also delivered fluorescently labeled siRNA into LFA-1 expressing K562 cells which were engrafted in the lungs of SCID mice after intravenous injection. A noncovalent ARC was described by using anti-prostate-specific membrane antigen (PSMA) scFv-protamine fusion complexed with siRNA specific for Notch1 [111]. The siRNA was efficiently delivered into PSMA-positive prostate cancer cells. The ARC significantly inhibited prostate cancer cell growth in vitro and in vivo. In addition to the antibody protamine fusion, other positively charged peptides and polymers were used for preparing ARC for siRNA delivery [107,110]. A CD7-specific antibody scFv containing C-terminal Cys was conjugated to an activated Cys containing oligo-9-Arg peptide, Cys (Npys)-(D-Arg)9 peptide [107]. The antibody-oligo-9-Arg peptide conjugate was then complexed with siRNA targeting CCR5 or CD4. The treatment of mice with the ARCs controlled viral replication and prevented the disease-associated CD4 T-cell loss after HIV infection. It also suppressed endogenous virus and restored CD4 T-cell counts in virus-infected mice reconstituted with PBMC. In another study, an aminooxy-derivatized cationic polymer was conjugated to anti-HER2 antibody through unnatural amino acid, pAcF, introduced at Q389 for full length IgG or S202 for Fab [110]. The antibody conjugates effectively and selectively delivered siRNAs and silenced multiple genes, including those encoding GAPDH, MDM2, and DNAJB11, in HER2-positive cancer cells in vitro.

### 5.2. Direct Conjugation of siRNA

Although the results are very encouraging by using highly positive charged protamine or other peptides/polymers in making noncovalent ARCs, there are still needs to improve the process. The direct conjugations of siRNA into specific sites in antibodies have been developed in silencing several genes in vitro and in vivo. Cuellar et al. first reported the direct coupling of antibody with siRNA using the THIOMAB^TM^ approach [112]. They selected antibodies, which are against seven different targets with differences in internalizations (such as those through lysosome, recycling, and slow internalizing), and introduced an unpaired Cys in either the heavy chain (A118C) or the light chain (V205C). These antibodies were coupled with 21-mer siRNAs (~15 kDa) specific for peptidlyprolyl isomerase B (siPPIB) and several other genes, generating conjugates with one or two (average 1.7 in ARC population) siRNA per antibody. Among the ARCs being tested in vitro, two displayed moderate gene silencing (≤50% silencing) and one (anti-TENB2-siRNA conjugate) with the highest silencing (>50%) in the cells expressing high antigen levels. The anti-TENB2 ARC also silenced PPIB mRNA expression by ~33% in the cells from mouse xenografts of cancer cells expressing high antigen levels after systemic administration. It was suggested that ARC entrapment in endocytic compartments is a limiting factor for silencing. In another study, Fab fragment with two thiol groups prepared from anti-CD71 (transferrin receptor) antibody was conjugated with maleimide containing siRNA with [113]. The ARC (1.2 to 2.2 siRNA per Fab) displayed gene-silencing in the heart and skeletal muscle after intravenous administration and in the gastrocnemius muscle when injected intramuscularly in mice. The treatment with myostatin-targeting ARC significantly silenced myostatin expression and hypertrophy of the gastrocnemius muscle after intramuscular injection in a mouse model of peripheral artery disease. NANOBODY^®^-siRNA conjugate has been described [114]. Anti-EGFR NANOBODY^®^ with an engineered C-terminal Cys were coupled to siRNA specific for a housekeeping gene (AHSA1) containing thiol-reactive groups. The ARCs maintained their EGFR binding and entered EGFR-positive cells via receptor-mediated endocytosis. They were active in vitro with significant reduction of mRNA expression by 70 to 98% in several targeting cell lines including A431, MDA-MB-468, and HepG2 cells. A different ARC was generated using engineered dual variable domain (DVD) antibodies [115]. As described in Section 2 on site-specific antibody conjugation using non-cytotoxic compound, a chemically programmable antibody Fv was engineered as inner Fv, which was derived from anti-hapten antibody and contained a uniquely reactive Lys residue at the bottom of a hydrophobic pocket with the hapten binding and catalytic site. The inner Fv selectively reacted with siRNAs derivatized with a β-lactam functionalized hepten group at the 3′ or 5′ of its sense strand. The same DVD antibodies also contained outer Fv that selectively targets cell surface antigens. The DVD-ARCs displayed effective targeting various cell surface antigens, such as BCMA, on multiple myeloma cells for the selective delivery of siRNA targeting β-catenin (CTNNB1). The in vitro treatment with BCMA targeting DVD-ARCs exhibited significant reduction of CTNNB1 mRNA and protein at concentrations as low as 10 nM.

### 5.3. Other Nucleic Acid

In addition to the direct conjugations which were validated for gene knockdown, other site-specific antibody conjugation approaches were developed using oligonucleotides without providing evidence of gene knockdown. Konc et al. reported a method for site-specific antibody-DNA conjugation using benzoylacrylic-labeled oligonucleotides [116]. The oligonucleotides were conjugated to antibodies or VHH molecules containing single engineering cysteine with high homogeneity. The superior performance of benzoylacrylic acid functionality over maleimides was observed in antibody-DNA conjugation reaction. The internalization of antibody or VHH conjugates was observed using HER2-positive cancer cells in vitro. An approach using template-directed covalent conjugation of DNA to antibodies was described [117]. A guiding DNA strand modified with a metal-binding functionality was directed to a second DNA strand containing N-hydroxysuccinimide to the vicinity of metal-binding site of the Fc domain histidine cluster of IgG1 antibodies in the presence of copper (II). The lysine residue close to the site was subsequently site-selectively conjugated. SpyCatcher-SpyTag system was applied for site-specific coupling of oligonucleotide to antibody [118]. SpyTag-oligonucleotide was shown to be conjugated to scFv fused with SpyCatcher on C-terminus through the formation of a covalent isopeptide bond. There is also a report on site-specific antibody conjugation with spherical nucleic acid (SNAs) [119]. The antibody-SNA conjugate was prepared by coupling SNA into unnatural amino acid introduced in the antibody using click chemistry.

Due to great therapeutic interest and progress made in related technologies, antibody-oligonucleotide conjugates have started to enter the clinic [120]. They were first advanced into a phase I trial of a rare muscle disease, followed by a few other biotech companies for many different indications including rare diseases, central nervous system disorders, and cancers. The oligonucleotides including siRNA and antisense oligonucleotide, which were coupled to antibodies, were used for gene knockdown or exon-skipping to induce protein translation from mutated genes. Although these initiatives are very encouraging, there are still challenges related to the antibody-oligonucleotide conjugations. They include the efficiency of cytosolic/nuclear delivery from endosomal escape, conjugation efficiency and yield, purification, and characterization due to large acidic payload [121]. Unlike synthetic cytotoxins used in ADCs which are often hydrophobic in nature and small in size (~2 kDa), the oligonucleotides are relatively large (>10 kDa) and hydrophilic which are less efficient in penetrate through the endosomal membrane. These properties of oligonucleotides also affect their process and characterization. Moreover, the presence of anti-nucleic acid antibodies in people with certain immune-mediated inflammatory disorder may have impact on safety, pharmacokinetics, and biodistribution of the conjugates. Nevertheless, site-specific antibody-oligonucleotide conjugations have created a great opportunity in therapeutics for many unmet medical needs in the future.

As described above in Section 3, Section 4 and Section 5, many different site-specific conjugation methods have been applied in generating antibody conjugates with various payloads other than conventional cytotoxic compounds or synthetic cytotoxins. They have advantages and drawbacks (Table 5).

Some of these methods, including Cys-mediated conjugation, have been used more frequently than others, probably due to relevant expertise, which individual researchers have, and no need for special reagents. A few methods seem to be used more often than others for particular kinds of payloads. For example, many conjugates containing non-cytotoxic synthetic compounds have been made by conjugation through Cys or unnatural amino acid, while enzyme modification with SrtA or FGE was often applied for coupling protein or peptide at C-terminus of antibody through short peptide tag. Since most of these payloads have different physico-chemical characteristics, their attachment to antibodies could result in different properties associated with the conjugates due to the interaction of payloads with nearby amino acid residues from antibody molecules. Thus, different conjugation strategies, methods and characterizations are required for utilizing individual payloads for robust process leading to stable and safe products. Nevertheless, knowledges and experiences will be gained with more of those modalities going through development in the future.

## 6. Conclusions

As a next generation method, site-specific antibody conjugation provides great potentials for producing homogenous ADCs with improved therapeutic index. Many novel methods have been developed with continued progress using antibody engineering approaches. They have been applied for coupling many different payloads, including non-cytotoxic compounds, small proteins/peptides, and siRNA, beyond synthetic cytotoxins for ADCs. These unique antibody conjugates may have potentials for developing new therapeutics against different diseases or providing proof of concept to evaluate novel concepts.

## Figures and Tables

**Figure 1 molecules-28-00917-f001:**
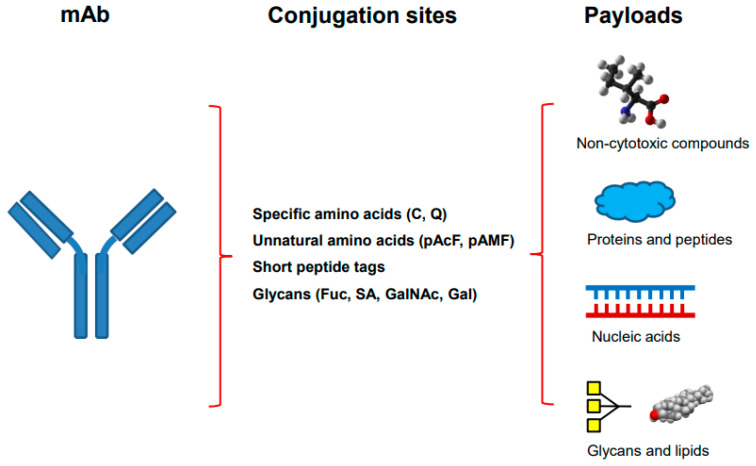
**The site-specific antibody conjugation using payloads other than synthetic cytotoxins.** The monoclonal antibody on the left is engineered by introducing different sites for selective coupling including specific amino acids, unnatural amino acids, short peptide tags, or modified glycans (middle). Different payloads, including non-cytotoxic compounds, proteins and peptides, nucleic acids, as well as glycans and lipids (right), are used for conjugation.

**Figure 2 molecules-28-00917-f002:**
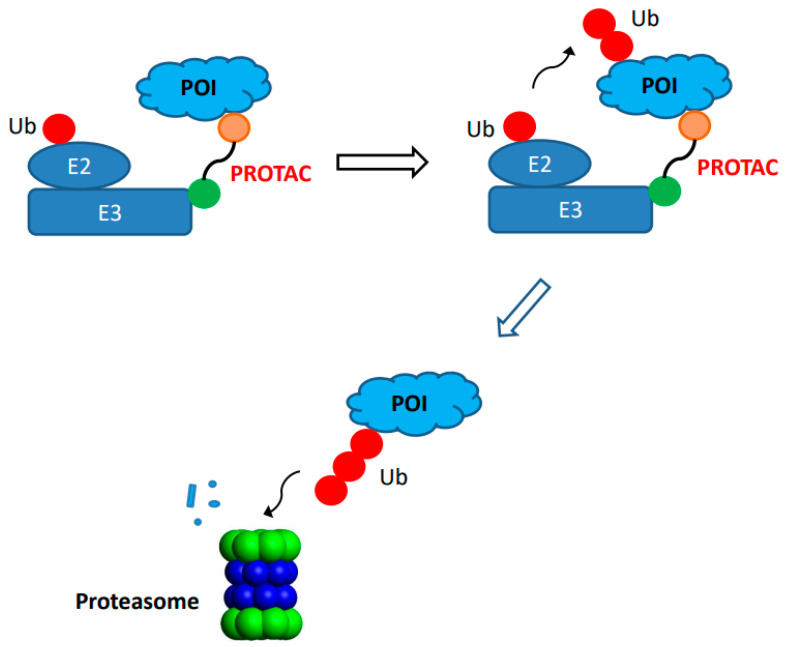
**Target protein degradation through proteolysis-targeting chimeras (PROTAC).** A bifunctional small molecule PROTAC compound is consist of ligand (circle in orange color) that binds the protein of interest (POI, light blue) and ligand (circle in green) for an E3 ubiquitin ligase (E3, rectangle in dark blue). Both ligands are connected by a linker. The binding of PROTAC compound to both POI and E3 ubiquitin ligase results in ternary complex formation, leading to addition of ubiquitin (circle in red) to POI that is degraded by proteasome inside the cells.

**Figure 3 molecules-28-00917-f003:**
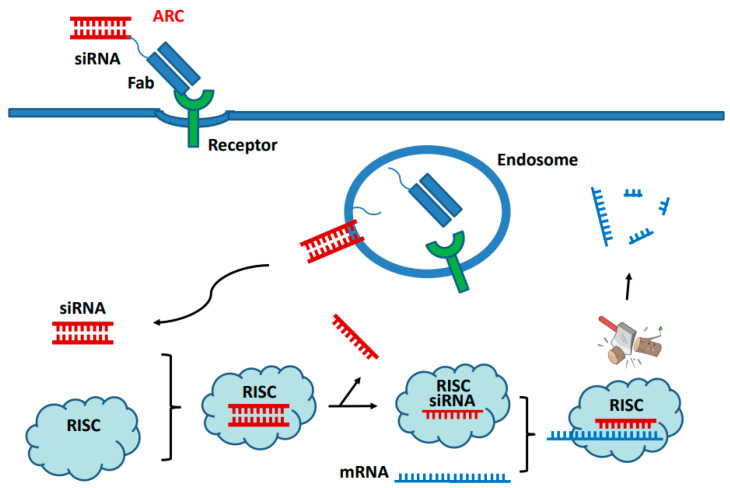
**The mechanism of action of antibody conjugated with short interfering RNA (siRNA).** Antibody–siRNA–conjugate (ARC) containing Fab and siRNA binds to the receptor on cell surface and is endocytosed into endosome inside the cells, where the siRNA is released and escaped into cytoplasm. The siRNA is then loaded into RISC, resulting in degradation of particular mRNA.

**Table 1 molecules-28-00917-t001:** The four categories of the site-specific antibody–drug conjugation.

Techniques	Conjugation Sites	Genetic Engineering	Metabolic Labeling	Chemo-Enzymatic Modification	Selective References
Specific amino acids	C (Cys), Q (Gln)	+	−	±	[14,15,16,17,18,19,20,21,22,23]
Unnatural amino acids	pAcF, pAMF, Sec, etc.	+	+	±	[24,25,26,27,28]
Glycans	Sialic acid, GalNAc, GlcNAc, Gal, Fuc, etc.	−	±	+	[29,30,31,32,33,34,35]
Short peptide tags	LL**Q**G, L**C**TPSR, etc.	+	−	+	[36,37,38,39,40,41,42]

**Table 2 molecules-28-00917-t002:** Site-specific antibody conjugations with non-cytotoxic compounds.

Categories of Payloads	MOA of Payload	Antibody Formats	Specific Site Used	Conjugation Chemistry	References
PEG	Prolonged serum half-life	Fab	Engineered Cys in C-terminus	Thiol-mediated conjugation	[57,58,59]
Antibiotics	Inhibitor of bacterial RNA polymerase	mAb	LC V205C	THIOMAB^TM^	[60,61]
Immune-modulating compounds	PDE4 inhibitor for immune suppression	mAb	Unnatural amino acid	Oxime chemistry	[62]
	Liver LXR agonist for immune suppression	mAb	Unnatural amino acid	Oxime chemistry	[63]
	Agonist of glucocorticoid receptor	Nb	C-terminal LPETGG	Sortase A (SrtA) mediated transpeptidation	[64]
Protein degraders	PROTAC-mediated ERa and BRD4 degradation	mAb	Engineered Cys	THIOMAB^TM^	[50]
	PROTAC-mediated BRD4 degradation	mAb	Hinge Cys	Click chemistry	[65]
		mAb	Engineered Cys	THIOMAB^TM^	[51,52,66]
	LYTAC-mediated degradation through ASGPR	mAb	FGly in C-terminus of HC, hinge, CH1	Hydrazino-iso-Pictet–Spangler reaction and click chemistry	[67]
	LYTAC-mediated degradation through M6PR	mAb	N-glycans	Chemoenzymatic reaction	[68]
Ligand for proteins overexpressed in cancer	Chemically programmed bispecific Fab as T-cell engager (Fab-synthetic ligands)	Fab	Unnatural amino acid at HC K138	Oxime chemistry	[69]
		Fab	C-terminal Sec in HC	SeH-maleimide chemistry	[70]

**Table 3 molecules-28-00917-t003:** Site-specific antibody conjugation with proteins and peptides.

Categories of Payloads	MOA of Payload	Antibody Formats	Specific Site Used	Conjugation Chemistry	References
Proteins	Enzyme	mAb	C-terminal FGly in HC	Oxime or Hydrazino-iso-Pictet–Spangler reaction and click chemistry	[88,89]
	Immunotoxin	mAb	M252 in Fc	Peptide-directed photo-cross-linking	[42]
	Cytosolic delivery through toxin	mAb	HC C-terminal LPSTGGK	Sortase A (SrtA)-mediated transpeptidation	[90]
Antibody fragments	Bispecific Fab as T cell engager (Fab-Fab)	Fab	Unnatural amino acid in LC or HC	Click chemistry	[91,92,93]
	Biparatopic scFv	scFv	C-terminus	SpyTag and SpyCatcher-mediated ligation	[94,95]
Peptides or cyclic peptides	Antimicrobial macrocyclic peptide (ABCs)	mAb	HC C-terminal (GS)_6_-LPETGGG	Sortase A (SrtA)-mediated transpeptidation	[96]
	Internalization through CPP	Nb	C-terminal LPETG	Sortase A (SrtA)-mediated transpeptidation	[97]
	Cytosolic delivery through cyclic CPP	Nb	C-terminus	Intein-mediated thioester (or expressed protein ligation)	[98]

**Table 4 molecules-28-00917-t004:** Site-specific antibody conjugations with siRNA.

Categories	Antibody Formats	Specific Site Used	Conjugation Chemistry	MOA of siRNA	References
Non-covalent complex with peptide or polymer	Fab, scFv	Protamine fused	Noncovalent complex	In vitro and in vivo gene knockdown	[106]
	scFv	Conjugated with Cys oligo-9 arginine	Noncovalent complex	Antiviral in vivo	[107]
	scFv	Protamine fused	Noncovalent complex	In vitro gene knockdown	[108]
	scFv	Protamine fused	Noncovalent complex	Antiviral in vivo	[109]
	Ab, Fab	Conjugated unnatural amino acid with cationic copolymer	Noncovalent complex	In vitro gene knockdown	[110]
	scFv	Protamine fused	Noncovalent complex	Anticancer	[111]
Direct conjugation	Ab	Cys	THIOMAB^TM^	In vitro and in vivo gene knockdown	[112]
	Fab	Cys	Thiol-mediated conjugation	In vivo gene knockdown in muscle	[113]
	Nb	Cys	Thiol-mediated conjugation	In vitro gene knockdown	[114]
	Ab	Engineered Lys	Lys-β-lactam	In vitro gene knockdown	[115]

**Table 5 molecules-28-00917-t005:** The advantages and disadvantages of major site-specific conjugation methods for payloads other than synthetic cytotoxins.

Methods Targeting Different Sites	Technical Categories	Advantages	Disadvantages
Cys-mediated conjugation	Specific amino acids	Well investigated No special reagent or enzyme required Broad range of DAR (2 to 6) Process scalable	Potential instability in vivo, depending on the site engineered with Cys
Conjugation through Gln295 in deglycosylated Ab using mTG	Specific amino acids	High stability High selectivity	Special enzyme required DAR limited to 2
Conjugation through introduced unnatural amino acid (pAcF)	Unnatural amino acid	High selectivity Process scalable Extensively investigated	Special reagent required Cell line engineering required
Modification (GlcNAc) using transglycosidase	Glycan	High stability High selectivity Applicable for high DAR (>2) No genetic engineering required	Special reagent and enzyme required
Modification using co-expressed FGE (Cys in LCTPSR)	Short peptide tag	High stability High selectivity	Immunogenicity in human unknown with introduced peptide
Conjugation using SrtA-mediated transpeptidation	Short peptide tag	High stability High selectivity	Special enzyme required Immunogenicity in human unknown with introduced peptide

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
