# Peer review of "Site-Specific Antibody Conjugation with Payloads beyond Cytotoxins"

_molecules, 2023, doi:10.3390/molecules28030917_

Round 1

Reviewer 1 Report

The current paper comprises the literature on antibody-drug conjugates. These antibody-drug conjugates are more specifically focused on protein degradation. Protein degradation, unlike classical inhibitors, which utilize occupancy-driven pharmacology, depends on event-driven pharmacology therefore, issues such emergence of resistance and continuous dosage are uncommon with these strategies. However, like other drug targeting strategies, these also have certain limitations, such as their molecular obesity restricting their crossing to the plasma membrane, and their clearance requires cellular metabolism. 

To an extent, antibody conjugation of drugs/medicinally-active compounds helps their endocytosis and improves their cellular selectivity as well.

There are certain points that, when added, will improve the readability of the paper.

1. Please write the advantages and disadvantages of site-specific conjugation, and try to compile them in table form.

2. Antibody-drug conjugate (Estrogen PROTAC): There are newer developments in this field because of the clinical success of ARV-PROTACs. Please add the following information on antibody-based estrogen PROTACs. (for more information: doi.org/10.3390/pharmaceutics14112523)

The paper is organized in a systematic way and fulfills all the requirements of a medicinal chemistry paper

Author Response

The current paper comprises the literature on antibody-drug conjugates. These antibody-drug conjugates are more specifically focused on protein degradation. Protein degradation, unlike classical inhibitors, which utilize occupancy-driven pharmacology, depends on event-driven pharmacology therefore, issues such emergence of resistance and continuous dosage are uncommon with these strategies. However, like other drug targeting strategies, these also have certain limitations, such as their molecular obesity restricting their crossing to the plasma membrane, and their clearance requires cellular metabolism.

To an extent, antibody conjugation of drugs/medicinally-active compounds helps their endocytosis and improves their cellular selectivity as well.

Response from author: I fully agree with the comment from the reviewer. The protein degraders use different mechanism of action (event-driven pharmacology) as compared to classical inhibitors (occupancy-driven pharmacology). This novel mechanism of action may provide new opportunity for cancer therapy, such as drug resistance, while the conjugation to antibody would facilitate selective tissue and cellular delivery.

There are certain points that, when added, will improve the readability of the paper.

  1. Please write the advantages and disadvantages of site-specific conjugation, and try to compile them in table form.

Response from author: The advantages and disadvantages of the site-specific conjugation methods have been added in lines 65 to 77 when these conjugations in general were discussed. A table (table 5) has also been prepared to highlight the benefits and limits of site-specific conjugations using the payloads other than synthetic cytotoxins (between lines 632 and 633). Related discussions have been provided following this table (lines 634 to 646). The author appreciates the valuable feedback from the reviewer.

  1. Antibody-drug conjugate (Estrogen PROTAC): There are newer developments in this field because of the clinical success of ARV-PROTACs. Please add the following information on antibody-based estrogen PROTACs. (for more information: doi.org/10.3390/pharmaceutics14112523)

Response from author: The information as well as the reference (#77) has been added as suggested by the reviewer (please see lines 254 to 255).

The paper is organized in a systematic way and fulfills all the requirements of a medicinal chemistry paper

Response from author: The author would like to thank the reviewer for great suggestions and helps to improve the quality of the manuscript.

Reviewer 2 Report

The manuscript reviews antibody conjugates with payloads, which do not represent direct cytotoxins, prepared via site-specific conjugation. The subject of the site-specific conjugation is a hot-spot in preparing drug conjugates for targeted therapy. An exclusion of direct cytotoxins may mean that an author does not deal with anti-cancer drugs. However, some of the payloads mentioned in the manuscript can be considered as having an indirect cytotoxic activity. In any case, the subject is of great interest and it is an author's merit that he (or she) has chosen this rather complex theme for reviewing.

1.The weakness of this review is in the absence of any analysis of presented results and points of view of other autors, whose papers are discussed. One could expect a comparison of different approaches to the preparation of antibody conjugates, or a discussion of fields, where conjugates of antibodies with non-cytotoxic payloads can be used, or a summary of various chemistries employed in conjugations, their advantages and drawbacks with regard to conjugate preparation and usage, as well as their interrelation with payload structures. The presented review manuscript after Line 106 looks like a collection of statements of paper abstracts from other abstracts with no analysis, discussion, criticism or summary of the presented material.

2. Since the author limits the payloads with non-cytotoxic compounds, an emphasis is needed to the differences in conjugation strategies, methods and/or means of use of conjugates with cytotoxic and non-cytotoxic payloads.

3. There are several very good reviews of conjugation methods describing their favorable properties and drawbacks (most of them are mentioned in the manuscript).  Taking into account that some reviews have been published in recent years, the author should emphasize new data and views that are not present in the previous reviews.   

4. "Bispecific Fab as T cell engager" are mentioned in tables 2 and 3; it would be better to show them in one and the same table.

5. Line 326: not "dibenzocycloocyte", but "dibenzocyclooctyne"       

Author Response

The manuscript reviews antibody conjugates with payloads, which do not represent direct cytotoxins, prepared via site-specific conjugation. The subject of the site-specific conjugation is a hot-spot in preparing drug conjugates for targeted therapy. An exclusion of direct cytotoxins may mean that an author does not deal with anti-cancer drugs. However, some of the payloads mentioned in the manuscript can be considered as having an indirect cytotoxic activity. In any case, the subject is of great interest and it is an author's merit that he (or she) has chosen this rather complex theme for reviewing.

Response from author: I really appreciate the comments from the reviewer. Although the site-specific conjugations using low-molecular weight cytotoxins have been covered extensively in literature with many excellent publications, the review on conjugation with payloads other than synthetic cytotoxins is limited. To clarify what are covered in this review, we defined the focus on payloads other than “synthetic cytotoxins” or cytotoxic compounds. The antibody conjugates containing these non-conventional payloads have also been demonstrated as potential therapeutics for cancer, immune-mediated inflammatory disorders, and infectious diseases. A few of them are currently under clinical trials.

1.The weakness of this review is in the absence of any analysis of presented results and points of view of other autors, whose papers are discussed. One could expect a comparison of different approaches to the preparation of antibody conjugates, or a discussion of fields, where conjugates of antibodies with non-cytotoxic payloads can be used, or a summary of various chemistries employed in conjugations, their advantages and drawbacks with regard to conjugate preparation and usage, as well as their interrelation with payload structures. The presented review manuscript after Line 106 looks like a collection of statements of paper abstracts from other abstracts with no analysis, discussion, criticism or summary of the presented material.

Response from author: There are analysis and points of view from the author in the end of each section 3 (lines 337 to 341), 4 (lines 473 to 478), and 5 (lines 608 to 625).  I also added more discussion based on the excellent feedback: lines 157 to 159; lines 190 to 192; lines 292 to 293; lines 309 to 310; lines 326 to 327; lines 373 to 375; lines 400 to 403; lines 446 to 450; and lines 621 to 624. Since most of abstracts from each work cited are quite concise, the author had to read each individual publication and collect relevant information from the text for this manuscript, such as what antibody engineering made for conjugation, conjugation chemistry and in vitro and in vivo study results. It is impossible to just put statements of these abstracts for writing this review paper.

The comparison of different site-specific antibody conjugations has been added in lines 65 to 77 in section 2 to provide summary of different site-specific conjugations and some of these methods have been used for coupling payloads other than synthetic cytotoxins. Since there are many excellent reviews on site-specific antibody conjugations in general, especially with cytotoxins, and it is not the focus for this manuscript, only brief discussion has been given. I hope that it is appropriate.

  1. Since the author limits the payloads with non-cytotoxic compounds, an emphasis is needed to the differences in conjugation strategies, methods and/or means of use of conjugates with cytotoxic and non-cytotoxic payloads.

Response from author: Thanks for great comment from the reviewer. A table (table 5) (lines 632 to 633) has been added to show advantages and disadvantages of different conjugation methods for using non-cytotoxic payloads. A follow-up discussion, such as the potential interaction of different payloads other than synthetic cytotoxins with amino acids from antibodies, has been included explain their properties in lines 640 to 646.

  1. There are several very good reviews of conjugation methods describing their favorable properties and drawbacks (most of them are mentioned in the manuscript).  Taking into account that some reviews have been published in recent years, the author should emphasize new data and views that are not present in the previous reviews.   

Response from author: I fully agree with this comment. Since there are many excellent reviews on different site-specific conjugation methods in general, the current manuscript only briefly discusses the advantages and disadvantages of different methods (lines 65 to 77). As suggested by the reviewer, the work from most of recent publications are highlighted here (lines 87 to 123). There is a sentence “Since progress in site-specific antibody conjugations has been reviewed in detail pre-viously [8-10, 34, 43-45], only the recent advances are highlighted here.” (lines 77 to 79) to emphasize new data being presented.

  1. "Bispecific Fab as T cell engager" are mentioned in tables 2 and 3; it would be better to show them in one and the same table.

Response from author: The “Bispecific Fab as T cell engager” as listed in tables 2 and 3 are different modalities although they have the same function as T cell engager. In table 2, it was prepared by using single anti-CD3 Fab (for interaction with T cells) conjugated with a synthetic compound as ligands for cancer associated proteins (“chemically programmed” as defined in literature). However, in table 3, it was generated by conjugation of two different Fabs (one is anti-CD3 for interaction with T cells, while the other is an anti-cancer associated proteins: there is no synthetic compound used). For clarification, I have changed the name listed in table 2 as “Chemically programmed bispecific Fab as T cell engager (Fab-synthetic ligands)” while the one mentioned in table 3 has been defined as “Fab-Fab”. A sentence was also added in lines 297 to 299 for further clarification of anti-CD3 conjugated with synthetic ligand.

  1. Line 326: not "dibenzocycloocyte", but "dibenzocyclooctyne"

Response from author: It has been corrected (currently in lines 368 to 369). Thanks.

Reviewer 3 Report

The Author of the review article “Site-specific antibody conjugation with payloads beyond cytotoxins” has described the advances in developing next-generation antibody conjugation methods. The progress in site-specific conjugation of various payloads (proteins/peptides, glycans, lipids, nucleic acids) to antibody molecules has been reviewed. The topic is interesting and may be of interest to many researchers. The manuscript is generally well-written and contains well-prepared schemes and valuable tables. In my opinion, the manuscript is suitable for publication after editorial correction.

Author Response

The Author of the review article “Site-specific antibody conjugation with payloads beyond cytotoxins” has described the advances in developing next-generation antibody conjugation methods. The progress in site-specific conjugation of various payloads (proteins/peptides, glycans, lipids, nucleic acids) to antibody molecules has been reviewed. The topic is interesting and may be of interest to many researchers. The manuscript is generally well-written and contains well-prepared schemes and valuable tables. In my opinion, the manuscript is suitable for publication after editorial correction.

Response from author: The author really appreciates the comment from the reviewer.

Reviewer 4 Report

In section 2, Overview of site-specific antibody conjugation, several recent studies of specific conjugations by various approaches were summarized and supplemented with Table 1, which is good. However, besides this laundry list, readers may expect to learn more insights, e.g. pros and cons of these approaches compared among them? 

Line 89, “In other report,” -> in other reports / in another report

In section 4, Proteins or peptides as payloads, the manuscript can be benefited from adding a discussion why conjugations instead of single fusion proteins are used – what the advantages of conjugations? 

In section 5, Nucleic acid as payloads: I understand this review paper focuses on ADCs, but a brief discussion on safety, PK, distribution, generation of anti-nucleic acid antibodies etc aspects can be a good addition for better overall understanding the idea of Ab-NA conjugations. 

Author Response

In section 2, Overview of site-specific antibody conjugation, several recent studies of specific conjugations by various approaches were summarized and supplemented with Table 1, which is good. However, besides this laundry list, readers may expect to learn more insights, e.g. pros and cons of these approaches compared among them? 

Response from author: Thanks for the suggestion. The pros and cons of different site-specific antibody conjugations have been added (lines 65 and 77). Furthermore, a table (table 5) (lines 632 to 633) has been included to compare different methods used for conjugation of payloads other than cytotoxic compounds, which are the focus of this manuscript.

Line 89, “In other report,” -> in other reports / in another report

Response from author: The words have been changed to “In another report” (now in line 104 after many additions). Thanks.

In section 4, Proteins or peptides as payloads, the manuscript can be benefited from adding a discussion why conjugations instead of single fusion proteins are used – what the advantages of conjugations?

Response from author: There is a paragraph to discuss the advantages of conjugation (lines 473 to 478). I also added another discussion for Fab-Fab conjugation (lines 446 to 450).

In section 5, Nucleic acid as payloads: I understand this review paper focuses on ADCs, but a brief discussion on safety, PK, distribution, generation of anti-nucleic acid antibodies etc aspects can be a good addition for better overall understanding the idea of Ab-NA conjugations. 

Response from author: A brief discussion has been included (lines 621 to 624).

Round 2

Reviewer 2 Report

The author has greatly improved the manuscript by adding some own notes and the new table 5. The manuscript can be accepted after a minor revision.

1. It is better to reconstruct the part of the text between the lines 67 and 123.  The part of the text starting from "THIOMAB ... " (line 82) should better be moved to the place after the phrase ending with "...Cys residue." (line 70). It will combine the description of using engineered Cys residues in a single place, while it is separated by the description of other approaches in the present version. The same is true for the description of the usage of unnatural amino acid residues as well as enzymatic modifications. 

2. Lines 67-67: "...clinical setting" - means "clinical trials"?  

3. Minor English editing is required.  

Author Response

The author has greatly improved the manuscript by adding some own notes and the new table 5. The manuscript can be accepted after a minor revision.

Response from author: I would like to thank the reviewer for his nice comment.

  1. It is better to reconstruct the part of the text between the lines 67 and 123.  The part of the text starting from "THIOMAB ... " (line 82) should better be moved to the place after the phrase ending with "...Cys residue." (line 70). It will combine the description of using engineered Cys residues in a single place, while it is separated by the description of other approaches in the present version. The same is true for the description of the usage of unnatural amino acid residues as well as enzymatic modifications. 

Response from author: The author really appreciates the valuable suggestion from the review. The part of the text between the lines 67 and 123 (currently in lines 66-121) has been reconstructed and I hope that it is good now.

  1. Lines 67-67: "...clinical setting" - means "clinical trials"?

Response from author: Yes. It has been changed to “clinical trials” for clarification (now in line 150-151)  

 Minor English editing is required.

Response from author: Minor English editing has been made as suggested by the reviewer, such as those in lines 10-16, 35-37, 60-63, 69-72, 235-236, 312-315, 504-509, 565-567, 581-583, 594-600, 625.